## COMMENT

# Consensus definitions of perception-action-integration in action control

Christian Frings [1,2,13✉], Christian Beste[3,13], Elena Benini [4,13], Malte Möller [5,13], David Dignath[6], Carina G. Giesen[7], Bernhard Hommel [8], Andrea Kiesel [9], Iring Koch[4], Wilfried Kunde[10], Susanne Mayr [5], Viola Mocke[10], Birte Moeller[1], Alexander Münchau[11], Juhi Parmar [12], Bernhard Pastötter [1,2], Roland Pfister [1,2], Andrea M. Philipp[4], Ruyi Qiu[5], Anna Render[5], Klaus Rothermund[12], Moritz Schiltenwolf [6] & Philip Schmalbrock[1]

The literature on action control is rife with differences in terminology. This consensus statement contributes shared definitions for perception-action integration concepts as informed by the framework of event coding.

## Main

In scientific communication, precise language and clearly outlined terms and definitions are of utmost importance. Precise terminology prevents misunderstandings and misconceptions, thus boosting scientific progress. Research in cognitive psychology has seen many highly paradigm-specific theoretical debates in their respective domain, which also resulted in paradigm-specific terminology. This is particularly true in the research area of action control. In general, action control describes how humans interact with their environment. Because actions are a hallmark output of the human cognitive system, the use of imprecise or inconsistent language to describe or explain phenomena within the domain of action control impedes scientific progress, particularly as regards unified theoretical approaches.

We present consensus definitions of central action control concepts from an event-coding perspective. Modern event-coding approaches including the Theory of Event-Coding (TEC[1]); or the Binding and Retrieval in Action Control framework (BRAC[2]); describe human action in an ideomotor context[3]. The basic assumption inherent in these approaches is that humans plan and execute actions through the anticipation of the perceptual effects of such actions. The anticipation (the mental representation) can retrieve motor patterns from memory necessary to ultimately achieve the anticipated effects. Accordingly, response, stimulus, and effect features can be represented together in feature compounds (or event-files, the central concept of these approaches). An event-file is an internal representation of characteristics of stimuli, responses, and effects elicited by the response, which can also be decoded in neural signals[4]. While object-files link perceptual features into coherent object representations (e.g.,

[1] Department of Cognitive Psychology, University of Trier, Trier, Germany. [2] Institute for Cognitive and Affective Neuroscience (ICAN), University of Trier, Trier, Germany. [3] Cognitive Neurophysiology, Department of Child and Adolescent Psychiatry, Faculty of Medicine, TU Dresden, Dresden, Germany. [4] Department of Cognitive and Experimental Psychology, University of Aachen, Aachen, Germany. [5] Department of Psychology and Human–Machine Interaction, University of Passau, Passau, Germany. [6] Department of Psychology, Eberhard Karls Universität, Tübingen, Germany. [7] Department of Psychology, Health and Medical University Erfurt, Erfurt, Germany. [8] Department of Psychology, Shandong Normal University, Jinan, Shandong Province, China. [9] Department of Psychology, University of Freiburg, Freiburg, Germany. [10] Department of Cognitive Psychology, University of Würzburg, Würzburg, Germany. [11] Institute of Systems Motor Science, Center of Brain, Behavior and Metabolism, University of Lübeck, Lübeck, Germany. [12] Department of General Psychology II, Friedrich Schiller University Jena, Jena, Germany. [13] These authors contributed equally: Christian Frings, Christian Beste, Elena Benini, Malte Möller. ✉email: chfrings@uni-trier.de

color, location, shape etc.), event-files add the response and effect component to the concept[5]. Event-files thus link perception and action and are conceptually similar to so-called *instances* in the Instance Theory of Automatization[6]. The consensus definitions we present here stem from approaches that are concerned with event-files (TEC) and how they are dynamically managed (BRAC).

Contemporary research in cognitive (neuro-)science on human action commonly employs so-called *action control paradigms* like, e.g., task switching, priming, and response-binding tasks. Event-coding approaches can describe results from these and further paradigms in terms of event-file binding and event-file retrieval due to one methodological aspect these entire tasks share: their sequential nature. In these tasks, participants respond to sequences of trials, a *prime* followed by a *probe*. Particular emphasis is given to how the characteristics of the prime trial *n* −*1* impacts behavior in the probe trial *n* (see Fig. 1). Often, the probe immediately follows the prime, but there is also research examining longer prime-probe intervals. In many paradigms, half of the trials function as the prime and the other half as the probe. However, in other paradigms, each trial comprises both prime and probe based on the specific pair of trials taken in consideration. In most paradigms, prime and probe trials require participants to perform an overt response (like a keypress) and behavioral effects are typically measured at the time of the probe only. The basic assumption of event-coding approaches is that stimuli, responses, and effects encountered in the prime are integrated (bound) into an event-file (Box 1). This event-file is

created after the prime, and decays in the time interval between prime and probe. If any feature is repeated at the time of the probe, the prime event-file is retrieved/reactivated and will influence probe responding. Specifically, retrieval facilitates performance when all the retrieved features match with the current features. However, retrieval hinders performance when the retrieved prime features are incompatible with the probe features (Box 2). Event-coding can thus unify paradigm-specific approaches in the action control literature as it provides a single account for a multitude of specific experimental effects. Figure 1 depicts the definitions provided in the current paper in relation to the typical stream of events occurring in an action control task. Of note, the presented consensus-definitions cannot only prove useful to researchers following an event-coding approach but also for all experimental approaches where sequential action control tasks are used[7]. The primary empirical evidence that we draw on for this framework and many details mentioned in these definitions can be found on OSF (https://osf.io/t6549/) for further reading.

### Limitations

The event-coding perspective on action control discussed here is mostly concerned with "simple actions" in the laboratory which, however, also occur in real world settings, (e.g., pressing a brake, typing, grasping objects). Complex cognitive functions like decision making, attitude formation, etc. are beyond the current scope of this approach. In addition, when event-coding is used to explain behavior outside the laboratory, it becomes

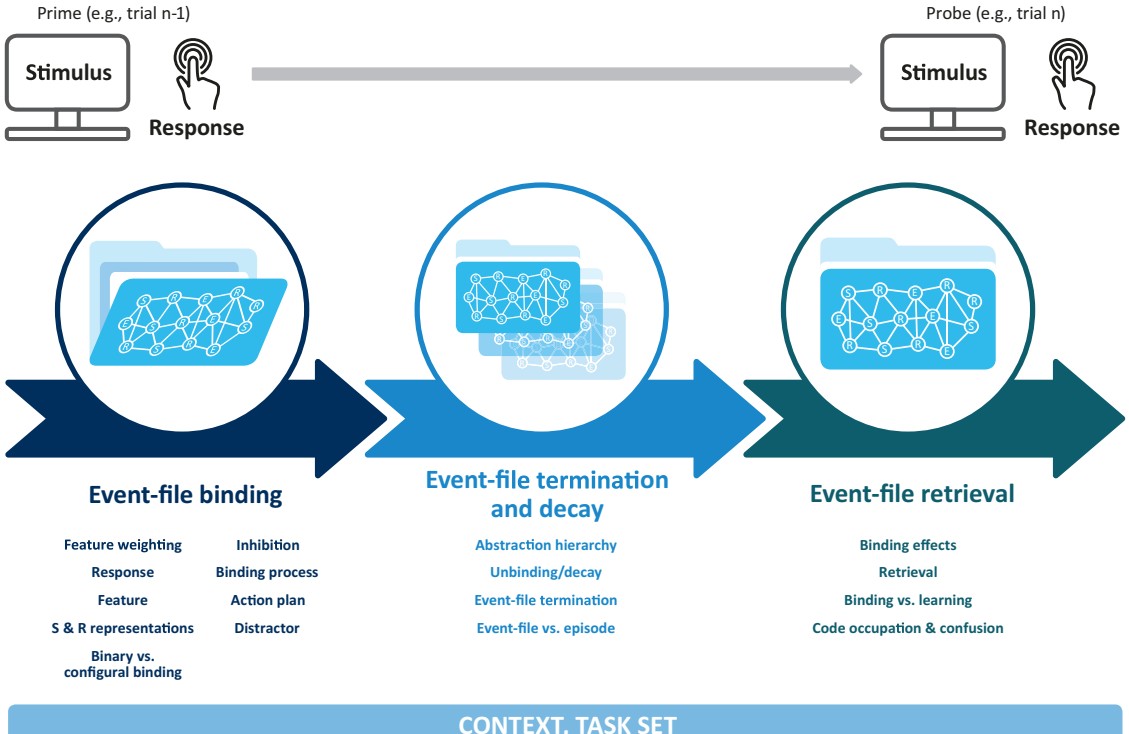

**Fig. 1 Typical sequential prime-probe structure of action control tasks together with the presumed processes of event-file binding, termination/decay, and retrieval in the stream of events.** The leftmost part of the figure represents the prime period, wherein participants give (or plan) a response to a stimulus. During the prime, the integration, or binding of an event-file occurs. The concept of an event-file is illustrated as a "folder" that can potentially contain different stimuli (S), responses (R) and response effects (E) representations, depicted as nodes on the "cover" of the folder. The folder is open, indicating that feature binding processes are ongoing at this stage. The central part represents the completion of the binding processes. At this stage, the event-file decays and/or its integrated features may be unbound. Finally, the rightmost part represents the probe trial, wherein participants respond to a stimulus and retrieval may occur if some stimuli, responses or effects are repeated from the prime. The closed folder depicts the prime event-file, the bound features of which are reactivated in the case of retrieval. All the concepts discussed in this terminology paper are reported at the bottom of the figure. The concepts are organized from left to right, based on where they originate or where they need to be discussed against alternatives.

---

**Box 1 | Event-file binding**

**Feature**

The term "feature" refers to perceptual attributes of stimuli (e.g., the location of a stimulus or the direction of a movement), but also to attributes of actions and their perceivable effects. Features of stimuli and actions are represented in the brain by means of feature codes. According to the ideomotor principle, actions can be represented in terms of their perceivable and anticipated perceptual effects. Therefore, the nature of feature codes representing stimuli is not distinct from feature codes representing actions. Action effects also include internal, affective consequences that become integrated into event-files and can also act as retrieval cues for previous event-files. In addition, features may also include more complex aspects of a stimulus or a response such as experiment-dependent semantic meaning. Imagine, for example, moving a joystick: it may indicate an approach/avoidance response in a certain setting or an upward/downward movement in another.

**Response**

A motor pattern that is cognitively represented by,and accessed through its anticipated perceptual consequences. Responses thus encompass the motor activity underlying them, as well as what agents perceive as effects of these activities. Moreover, responses can be represented based on different features, for example, in terms of body-related features (e.g. moving the left or right hand), and/or in terms of their semantic features (e.g. approaching or avoiding an object). Often, such features are not exclusive and responses are thus represented as compounds of feature codes. Responses that are executed in close temporal contiguity can also become bound to each other into episodic compounds, which is then referred to as response-response bindings stored in action files.

**Distractor (vs. irrelevant object/feature)**

A distractor refers to a stimulus or feature that is presented in addition to task-relevant stimuli/features. Importantly, distractors have a response assigned to them, which might differ from the response required in the current situation. In contrast, irrelevant features or irrelevant objects are also task-irrelevant, but do not have a response assigned to them. Both distractors and task-irrelevant features might interfere with response selection. Distractors, though, are likely to exert a stronger interference due to the response they are assigned to. Both distractors and task-irrelevant features can be integrated in an event-file, and can trigger retrieval of such event-file later on.

**Feature weighting**

Feature weighting refers to selective processing of features whereby some features receive more or prioritized processing. Feature codes with higher weights are more likely to be bound into an S-R episode and more likely to trigger retrieval. These weights are modulated by top-down factors, such as attention and task relevance, as well as bottom-up factors, such as salience.

**Binding**

Binding (as a process) refers to the formation of temporary links between two (or more) feature codes, which can refer to features of the stimulus, the response as well as response effects features. Binding (as a product) refers to the outcome of this integration process, i.e., a transient episodic representation of integrated (sensorimotor) feature codes commonly referred to as an "event-file."

**Binary and configural bindings**

Binding is considered "binary" if it consists of a link between two feature codes, while a configural binding consists of a link between three or more features. The distinction between binary and configural bindings has theoretical implications: repeating a feature of a configural binding retrieves the entire event-file. Therefore, performance benefits will occur only when all the retrieved features match the current features. Existing evidence tentatively suggests that the ease of decomposing complex stimuli into individual features, feature variability, as well as feature saliency increase the probability of forming configural rather than binary bindings.

**Action plan**

An action plan is an event file comprising integrated features of intended body-external and body-related perceptual changes, to be brought about by the corresponding efferent activity. An action plan is a representation that precedes every goal-directed action. It does not have to be executed immediately, but can be prepared and retained for later execution.

**Inhibition**

Two major classes of the concept "inhibition" can be distinguished: "behavioral inhibition" and "cognitive inhibition." First, at the behavioral level, inhibition indicates the deactivation of a response representation to prevent responding at a given time (action postponing, action restraint/withholding, action cancellation/stopping). Second, at the cognitive level, inhibition refers to the suppression or attenuation of an internal representation (memories, thoughts, perceptions, emotions). Inhibition is often discussed together with (reduced) feature weighting or "handling" of distractors, i.e. weakening their influence.

---

clear that it is harder to define what constitutes an "event," e.g., when a simple action is a step in a sequence to reach an overarching goal. In such situations, it seems to be promising to relate event-coding to other approaches like event-segmentation[8]. Finally, event-coding approaches are over-arching frameworks describing human action beyond the scope of a particular paradigm. Yet, there are alternatives. For instance, another overarching approach is predictive coding[9]. Predictive coding refers to a theoretical framework that explains how the brain processes and interprets sensory information. It suggests that the brain generates predictions about incoming sensory inputs based on prior knowledge and expectations, and then compares these predictions with the actual sensory signals. Any discrepancies between the predictions and the actual inputs are used to update the internal models and refine future predictions. The same principle of prediction error minimization has also been used to provide an account of behavior[10], in which motor actions are not commands but descending

proprioceptive predictions. Yet, concerning the comparison to TEC/BRAC one might argue that predictive coding focuses on the top-down processing and generation of predictions, while binding and retrieval processes are more concerned with the bottom-up integration of sensory information and the retrieval of information. Predictive coding and processes as described by TEC/BRAC are not mutually exclusive and might interact with each other.

## Open questions

At the time of this writing, it is still unclear when the formation of an event-file starts and when it ends. Binding processes can be expected at the time of response execution or the completion of an action plan, but partial repetition costs have also been observed for unpredictable effects and the latest response, and also for subsequent responses that were only planned after execution of the last. Furthermore, further characteristics of the

---

**Box 2 | Event-file retrieval**

**Retrieval**

Retrieval refers to the process of reactivating all feature codes stored in an event-file whenever one or more prime features or actions reoccur in the probe. Such a retrieval process may be comparable to retrieval triggered by a memory item-recognition test, in which the probe item is compared in parallel with all the stored items. Retrieval activates the respective features, thus facilitating their processing and the selection of responses that match with those retrieved. Retrieving an event-file yields measurable performance effects that are taken as evidence for the occurrence of binding during prime processing. Thus, binding is measured indirectly through the retrieval effects. Importantly, binding and retrieval are considered distinct processes, but retrieval is necessary to measure binding effects.

**Binding effects**

Binding effects are modulations of performance (in terms of e.g., speed, accuracy or choice behavior) attributed to feature binding and retrieval. Specifically, binding effects refer to impaired performance when some, but not all features of an event-file repeat between successive presentations (so-called partial repetition costs) as compared with either full repetitions or full changes. Binding effects are statistically indicated by an interaction of experimental factors, such as feature repetition versus change, between successive prime-probe presentations. In action planning paradigms, "partial repetitions conditions" are often labeled "partial overlap conditions" while binding effects are commonly termed "partial overlap costs." Binding effects in choice behavior refer to increased choice repetitions in the probe, when a prime feature repeats.

**Code occupation and code confusion**

The so-called code occupation hypothesis assumes that binding feature codes into an action plan temporarily prevents them from being included in another event-file. It has been proposed that creating an event-file with an already bound feature requires time-consuming disintegration (or unbinding) of the event-file and subsequent rebinding of the feature. Code confusion in turn attributes binding effects to a conflict that emerges when a retrieved event-file contains features that do not match with the current processing or task requirements, for example when repeating a stimulus feature retrieves a previously executed response that is incorrect in the current presentation. Thus, binding effects may be the result of additional processes (e.g., inhibition) needed to overcome conflicts arising from code confusion. However, it could be argued that these considerations are not mutually exclusive but may reflect different cases of conflict between feature codes: Both accounts concur that the need to represent two or more events with overlapping features creates conflict. If an already existing event-file (temporarily or permanently) hampers the creation of a new one, the code occupation account provides a more suitable characterization. If the newer, still-to-be-created binding is hampered by the existence of the older binding but eventually determines performance, the code confusion account provides a better explanation.

**Abstraction hierarchy**

Features can be represented on different levels of abstraction based on the experimental setting and its demands. For example, a color might be bound to a geometrical shape in a simple visual-search setting, or an imagined color might be bound to a location. In the same vein, both a motor response and a semantic classification can be bound to a stimulus, a response can be bound to a task-set, a stimulus or its physical dimensions might be bound to a cognitive control state such as a high versus low weight assigned to the distractor or task repetition versus task switch. Furthermore, event-files might become part of super-ordinate event-files that contain two (but possibly more) subordinated event-files.

**Episodes**

The term "episode," in contrast to event-files, refers to processes with an emphasis on their occurrence in time, integration (of stimulus and response codes) is the conceptual focus when referring to event-files, which is not necessarily happening in every episode. Thus, episode as a concept from the memory literature (the "classic episodic memory") is a little broader than an event-file (that emphasizes the link between action and perception).

**Unbinding/decay**

Event-files are found to have a limited lifespan during which they affect performance, which has been attributed to the decay of event-file representations over time. While some S-R episodes presumably can survive several minutes, hours, days, or months, thus serving a long-term memory function, other bindings seem to exert effects for a much shorter time.

In contrast, the term "unbinding" refers to the active disintegration of links between feature codes and the weakening of features (i.e. by inhibition) during reconfiguration upon reencountering an event-file. Precisely, how bindings disintegrate (vanish by time) remains an empirical question. Their activation may decay, so that bindings remain but no longer affect behavior, or they may be inhibited through the activation and/or retrieval of competing bindings (i.e. through interference).

---

environment (e.g., changes in context or effectors) might attenuate bindings between sequential stimuli, responses, or events, likely due to segmentation (Box 3).

In addition, it is a crucial question how transient bindings relate to more enduring and longer lasting learning effects. While some authors use the terms "binding" and "learning" interchangeably, others argue that binding and learning can be disentangled, and even suggest that binding products can be seen as building blocks of learning. Learning may reflect more consolidated representations of event-files. Given the scarcity and inconsistency of empirical findings on the relation of binding and learning more systematic research is needed that incorporates known modulators of learning in the action control literature[11].

## Conclusion and outlook

The goal of this paper is to reduce confusion about these terms and definitions in the field that may also be relevant to counteract the "replication crisis." The latter has been attributed to emerge from underpowered studies, publication bias, problems with applying statistical procedures, as well as imprecise theories[12] and/or misunderstandings and uncertainties to dealing with terms in such theories[13]. Therefore, having a clear basis of communication will be of help to counteract replicability problems. In addition, it will facilitate efforts, such as the ManyLabs initiative in psychological and neuroscience fields, in which different laboratories (with scientists from different professions as well as terms and definitions) work together. Recently, it has been outlined that effects of efforts related to replications and pre-registrations to counteract the replication crisis are limited because they cannot overcome problems that refer to the "base rate" of phenomena; i.e. the probability that a sought-after effect is truly present in the population[14]. Clearly, when communities use similar terms for different phenomena that are focused in their research (linked to base rates in the population), replicability must be low and scientific progress slow. We therefore think that this article will advance the field providing a better common ground in terms and definitions to be used in future studies and focusing on basic principles of how perception and action become integrated during action control.

---

**Box 3 ▐ Task set and context from an event-coding perspective**

**Task set**

A task set comprises the mental representations of those stimulus dimensions that are relevant for the current task, together with the response options and stimulus-response mappings. Moreover, it comprises the cognitive control processes that enable the translation of current perceptual information and task demands into appropriate motor output. In action control research, the task set therefore determines the rules and the boundaries of the goal-directed behaviors participants engage in. In multitasking research, investigating how agents deal with multiple tasks, experimental settings are thought to comprise multiple task sets. Task sets can have components that can be either "modular" (i.e., can be reconfigured independently) or "integrated" (can be changed only as a whole).

**Context**

The term "context" is broad and can refer to any aspect of the environment including, for instance, stimulus features and background noise, spatial locations where stimuli appear, or states of the person, e.g. affect. Such a broad definition clearly limits the possibility to truly define context boundaries. However, action control research has also adopted a narrower definition, whereby the context is a task-irrelevant event that may not be constant throughout a stream of events and can alternate as the task-relevant features do. Accordingly, contextual features are nominally irrelevant since they are not part of any task set, that is, they neither determine the rules nor the boundaries of a goal-directed behavior. Nonetheless, context features (of many kinds, see above) can be integrated into event-files and then serve as retrieval cue for this event-file. Alternatively, context representations may modulate binding processes by inducing event segmentation. Namely, a context switch sets a boundary that dissects previous series of events into a coherent representation matching the expectations of how a series of events usually unfolds. Switching the context segments the current trial from the previous trial, while a context repetition might not set such a boundary so that the different events are considered as belonging together. Hence, the context would not work as a retrieval factor, but would reduce activation of the previous-trial features in case of a switch versus leaving it unaffected in case of a repetition.

---

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

## Acknowledgements

The Deutsche Forschungsgemeinschaft (DFG) supported the research reported in this article (FOR 2790 and FOR 2698). The DFG had no role in preparation of the manuscript and the decision to publish.

## Author contributions

C.F., C.B., E.B., and M.M.: conceptualization, resources, writing—original draft, writing—review and editing, supervision. D.D., C.G.G., B.H., A.K., I.K., W.K., S.M., V.M., B.M., A.M., J.P., B.P., R.P., A.M.P., R.Q., A.R., K.R., M.S., P.S.: conceptualization, writing—original draft.

## Funding

## Competing interests

The authors declare no competing interests.
