## [Peer review file · Communications Psychology]

1st Sep 23

Dear Christian,

Thank you for your patience during the editorial evaluation and peer-review process.

Your manuscript titled "Consensus definitions of Perception-Action-Integration in Action Control" has now been seen by 3 reviewers, and I include their comments at the end of this message.

The reviewers are enthusiastic about your work and I share their positive views towards the future utility of the work for researchers within the TEC/BRAC domain, as well as those who work in the wider area.

We are very interested in publishing your Comment in Communications Psychology, but you will need to undertake some substantive revisions to ensure the piece fulfils its potential and complies with our Comment format.

With regard to the reviewer feedback, we ask you to prioritize the following three key issues:

- 1) There is currently no clear and consistent distinction between the individual definitions and the wider theory (and between task(s) and theory).
- 2) Some definitions are unclear or seem contradictory.
- 3) The text needs to be accessible to readers from outside the field, which is conceptually encapsulated as Reviewers #1 and #3 highlight.

These reviewer requests should be incorporated as part of a substantive revision that engages with the following issues:

Conceptually, your text can be considered as making two key contributions: 1) offering a set of definitions on which some leaders of the field agree, and 2) engaging in a critical discussion regarding open questions and reasons for the lack of agreement in some domains. While the definitions are mostly present (though see the referees' detailed comments), many conflate providing an accessible and easy-to-comprehend statement with critical discussion and expressions of ambiguity.

The Comment is also more than 4000 words long (the limit for Comments is 1600 - 1800 words), and the referencing style is unsuitable for the format (Comments have up to 10 references, which we can extend to 15 as a firm upper limit).

In revision, we ask you for the following changes to address all of these concerns:

- 1) Please ensure the definitions are just that - clear descriptions of terminology that are agreed upon - and placed inside "Boxes". I recommend the use of 3 boxes that are thematically structured. I left some suggestion on the attached version of your manuscript. The text in the boxes will not count towards the overall word limit, but for the sake of readability, definitions should be concise. Also for the sake of readability, please refrain from any cross-referencing between individual definitions: the list is so short that readers can locate information unaided.
- 2) The remaining text should be about 1200 words (1500 max) and critically engage with the state of the field, as you see it in light of the task to create a common glossary and taking into account the reviewers' comments. Concepts for which no consensual definition can be derived would better be discussed critically as open questions than presented with ambiguous definitions in the "boxes". The

distinction between definitions and theory that the referees asked for will be aligned with the distinction between information offered in the boxes and discussion in the main text. Please use informative subheadings to structure the text.

3) Comments should only reference key sources that the readers need to understand a controversy or for critical further reading. Self-references (to any contributing author) should be entirely avoided. In particular, because the present manuscript is an opinion piece, there is limited scientific value in referencing the same opinion expressed previously by some of the same authors elsewhere.

In sum, we invite you to revise your manuscript taking into account all reviewer and editor comments.

To aid you with that task, I have included a marked-up version of your manuscript.

EDITORIAL POLICIES AND FORMATTING

You will find a complete list of formatting requirements following this link:

<https://www.nature.com/documents/commsj-style-formatting-checklist-review-perspective.pdf>

Please use the checklist to prepare your manuscript for resubmission.

If you have any questions about any of our policies or formatting, please don't hesitate to contact me.

Please use the following link to submit your revised manuscript and a point-by-point response to the referees' comments (which should be in a separate document to any cover letter):

[Link redacted]

We hope to receive your revised paper within 4 weeks; please let us know if you aren't able to submit it within this time so that we can discuss how best to proceed. If we don't hear from you, and the revision process takes significantly longer, we may close your file.

We understand that due to the current global situation, the time required for revision may be longer than usual. We would appreciate it if you could keep us informed about an estimated timescale for resubmission, to facilitate our planning. Of course, if you are unable to estimate, we are happy to accommodate necessary extensions nevertheless.

Please do not hesitate to contact me if you have any questions or would like to discuss these revisions further. We look forward to seeing the revised manuscript and thank you for the opportunity to review your work.

Best wishes,
Marika

Marika Schiffer, PhD
Chief Editor
Communications Psychology

REVIEWERS' COMMENTS:

Reviewer #1 (Remarks to the Author):

Frings and colleagues present proposed consensus definitions of commonly used terms in the "theory of event coding" research literature. This sort of exercise can certainly be useful for promoting progress in the field, and the paper is generally scholarly and well written. For a wider audience to be able to place this effort accurately, however, a bit of additional tightening (and elaboration) of some of the definitions would be in order.

My major point is that I think the authors need to be more careful and accurate in delineating the goals and target literature (and audience) of this paper. Specifically, this set of definitions concerns a very particular literature within a (mostly Germany-based) ideomotor theory approach to event coding and action selection. This is not to say that it is not important or valuable, but it would be crucial to be explicit about the relatively limited scope of the paper's goals, so as not to give the naïve reader the impression that this paper reviews terminology concerning the control of actions more generally (which would be a vast literature, most of which outside of the ideomotor theory perspective). For example, rather than referring to "action control" per se, it would be more accurate to provide a qualifier that delineates the specific research niche the authors are operating in here, such as "an event coding-based perspective on action control" or similar. Some of the current wording and definitions provided give the impression of much broader research literature coverage, which I believe is simply not quite accurate. Consider, for example, the definition of action control provided on lines 53-55: "action control describes how humans interact with their environment, how they translate goals into actions, and how they relate perceptions to actions and vice versa". This definition would cover, for example, the entire literatures on decision making, motor control, and executive function (and more), but most researchers in those literatures likely have never heard of the notion of "event file binding", etc. To illustrate: you could read through entire editions of the current bestselling (typically US-based) cognition and cognitive neuroscience student textbooks without coming across the term "event file" even once, even though these books do of course cover the topic of "action control" in many different ways. It is therefore important to be more precise and explicit in informing the reader about the very specific field/perspective that these definitions concern. There are thousands of researchers working on understanding the control of actions who do not share the assumptions laid out in this article, so the language used should be more specific and qualified.

Minor points:

In the definition of "event file", the factor of time should be mentioned, in that it is the simultaneous or near-simultaneous experience of different stimulus features, etc. that results in them being bound together. Relatedly, I think a more explicit delineation of the current definition of "events" and the literature on event cognition (or event segmentation) would be useful somewhere in the paper. The latter literature is more concerned with temporally extended (real life) events, whereas the event coding literature seems to be focused on instantaneous, "trial-like" events exclusively (which is arguably further removed also from the everyday usage of "event", so this might also be worth pointing out early on).

In the same section, the distinction drawn between Logan's instance theory and event coding is not very clear, please elaborate some more. (Specifically, "one even file each" sounds a lot like "one memory trace per instance").

On line 169, the authors use (I believe for the first time) the term "feature code", and later they discuss things like "code occupation". The term "code", as used in this literature, also needs a definition, in my view.

The "feature weighting" section seems to describe an attentional function (weighing task-relevant features more than irrelevant ones) without using the term "attention". What is the relationship between feature weighting and attention?

In the action plan section, it is stated that an action plan precedes *every* action. This sounds odd when applied to completely reflexive or highly overlearned reactions. Is it really a "plan" when I pull away my hand from a hot stove plate?

On lines 381-383 the authors bring in the notion of event segmentation, but I don't think whatever point they are trying to make here is explained sufficiently.

Line 388: "...as one brick...". I had trouble understanding this sentence - what does this mean? Is this meant to refer to a "building block"? Please clarify!

Reviewer #2 (Remarks to the Author):

Frings et al. present a short article attempting to provide consensus definitions of important terms in the action control field. The article is quite clear and will be of use to those in the field. It is highly focused on the action control field, and therefore may be of less interest to those outside the field however I believe it will still provide a useful contribution to the literature. I do have a few comments that the authors might find useful:

1. The brevity of the article sometimes means the authors presuppose knowledge, and therefore the writing can be slightly confusing. For example, in the "feature weightings" section you state "Such weights, in turn, can influence effects of code confusion in partial repetition effects". Sentences like this (of which there are several) might be better unpacked and explained a bit more clearly.
2. In the "binary vs configural bindings" section, it is not clear to me whether there is a difference between multiple binary bindings vs a single configural binding. Are these two possibilities distinguishable?
3. In the "action plan" section it wasn't clear to me whether an "action plan" was always online - an active representation created just before the action occurs - or whether it can also be stored/retained (i.e., within an event-file). A bit more background information/examples would help here.
4. Sometimes your "definitions" are more about the wider theory than a clear definition. For example, in the "event-file termination" section I don't think you actually define the term. It might be obvious, but given this article is about definitions this still seems important. A broader point is also perhaps the need to make clear what isn't included in each definition - how has the term been used differently in the past which you think was incorrect? This latter point might further help clear up confusion in the

literature.

5. In the "event-file vs episode" section you link to episodic memory (as defined by Tulving), but don't discuss the "event segmentation" literature (e.g., Kurby & Zacks, 2008, TiCS). This seems relevant, given they specifically use the term "events" and it is more related to online processing of information than the episodic memory literature.

5. In the "retrieval" section, and earlier in the paper, you say that event-files are 'retrieved'. How is this distinguishable from continued activation of an event-file from the prime trial in the case of an immediate probe? Is retrieval always thought to occur rather than simple continued activation/maintenance?

Reviewer #3 (Remarks to the Author):

This article by Fringe and colleagues is essentially a glossary of concepts that arise in the context of ideomotor-based theories of action control. As someone not initially well-versed in these specific theories (for context, my research is focused on behavioral investigation and computational modeling of motor control, motor learning and cognitive control), I did find the article helpful to better understand the theory and current issues better. That said, it's certainly not a self-contained introduction to the theory. There is very little discussion of the evidence supporting these theories.

I am sympathetic to the importance of ensuring that all researchers reach a consensus on what different terms mean. A lot of unnecessary confusion and disagreement can arise from inconsistent terminology, leading to wasted effort, particularly when trying to bridge across research areas. With that in mind, I do have some comments about the terminology and definitions provided here, which I feel might end up creating barriers to other researchers engaging with these theories.

The main issue I have is that the terminology seems to blur the boundary between describing the phenomena that the theory seeks to explain, and describing the theory itself. Right from the beginning, on line 152, a "Response" (second definition) is defined as "a motor pattern that is cognitively represented by, and accessed through, its anticipated perceptual consequences." So, immediately, the definition here of what should be a very simple term ("response") – used for decades throughout neuroscience and psychology – has the ideomotor theory embedded into it. What would a motor pattern that is NOT represented in terms of its perceptual consequences be called? Do the authors believe that no such thing is possible? This creates exactly the kind of conceptual barrier across sub-disciplines that this article is seeking to overcome. A reader not already subscribed to the ideomotor principle will be alienated already.

There are more examples where this kind of issue arises. Another one is on line 169 - "a stimulus might instruct a specific response or can be the outcome of a response". This is fundamentally at odds with how the term "stimulus" is used elsewhere in neuroscience and psychology. If you are bought into the ideomotor principle, then this statement can make some sense. But I suspect this kind of definition, which again embeds the ideomotor principle is being embedded into the definition of terms, will create a barrier to integrating these ideas with other research areas.

A somewhat related issue is on line 305: "Binding effects in performance refer to longer response times in partial repetition conditions compared to either full repetitions or full change or both." My problem here is that a straightforward empirical phenomenon (differences in reaction time) is described by the putative underlying mechanism (binding). This will likely result in confusion (or worse, dogma) about how such effects can be explained.

In general, I would recommend that separate terminology should be used to describe tasks/empirical phenomena versus possible mechanisms/interpretations. If the terminology used to describe the experimental paradigms and empirical phenomena cannot be disentangled from the theory that explains them then that is problematic. I suspect this type of issue will prove to be an obstacle to the insights and ideas here being appreciated widely and incorporated into a broader research program. I should emphasize that I think this are widespread issues in neuroscience/psychology (e.g. tasks like Stroop/Simon are often described as "inhibition tasks" – a putative mechanisms).

I want to emphasize that I offer these comments in the spirit of trying to help support the authors' initiative to improve consensus and sharing of ideas across sub-disciplines. I find the underlying ideas interesting and I sincerely hope that we can make progress towards overcoming the siloed nature of a lot of research (especially in the case of action control).

Finally, on a different note, I do have some concerns about the extent to which the BRAC theory can be falsified. Later sections of the paper refer to a number of effects that are difficult to reconcile with the basic theory, but these are invariably explained away by amendments/extensions to the core theory. There is no discussion of alternative theories or perspectives. On that note, I wonder if some of the effects could be understood in terms of normative (e.g Bayesian) principles, rather than an ever-more-complex event-file mechanism? Note that such an approach (if it's possible; I have no idea) wouldn't be mutually exclusive to a mechanistic account, but might provide an alternative/complementary point of view through which some phenomena may be more easily explained.

University of Trier · 54286 Trier

Marike Schiffer, PhD

Chief Editor Communications

Psychology

FB I-Cognitive Psychology
Prof. Dr. Christian Frings

Universitätsring 15
54296 Trier
Phone +49 651 201-2958
chfrings@uni-trier.de
www.uni-trier.de

CF/uw
2023-10-30

POINT-BY-POINT RESPONSE LETTER

Comments to points raised by Reviewer 1

REVIEWERS' COMMENTS:

Reviewer #1 (Remarks to the Author):

Frings and colleagues present proposed consensus definitions of commonly used terms in the “theory of event coding” research literature. This sort of exercise can certainly be useful for promoting progress in the field, and the paper is generally scholarly and well written. For a wider audience to be able to place this effort accurately, however, a bit of additional tightening (and elaboration) of some of the definitions would be in order.

RESPONSE: Thank you for this positive feedback!

My major point is that I think the authors need to be more careful and accurate in delineating the goals and target literature (and audience) of this paper. Specifically, this set of definitions concerns a very particular literature within a (mostly Germany-based) ideomotor theory approach to event coding and action selection. This is not to say that it is not important or valuable, but it would be crucial to be explicit about the relatively limited scope of the paper’s goals, so as not to give the naïve reader the impression that this paper reviews terminology concerning the control of actions more generally (which would be a vast literature, most of which outside of the ideomotor theory perspective). For example, rather than referring to “action control” per se, it would be more accurate to provide a qualifier that delineates the specific research niche the authors are operating in here, such as “an event coding-based perspective on action control” or similar. Some of the current wording and definitions provided give the impression of much broader research literature coverage, which I believe is simply not quite accurate. Consider, for example, the definition of action control provided on lines 53-55: “action control describes how humans interact with their environment, how they

translate goals into actions, and how they relate perceptions to actions and vice versa". This definition would cover, for example, the entire literatures on decision making, motor control, and executive function (and more), but most researchers in those literatures likely have never heard of the notion of "event file binding", etc. To illustrate: you could read through entire editions of the current bestselling (typically US-based) cognition and cognitive neuroscience student textbooks without coming across the term "event file" even once, even though these books do of course cover the topic of "action control" in many different ways. It is therefore important to be more precise and explicit in informing the reader about the very specific field/perspective that these definitions concern. There are thousands of researchers working on understanding the control of actions who do not share the assumptions laid out in this article, so the language used should be more specific and qualified.

RESPONSE: The reviewer is right that there are many different ways of looking at 'action'. It is also true that many US-based textbooks are not particularly prone to covering this material. We hope to change this state of affairs, partially with this terminology paper, because the concept of event files has proven useful across different fields. It also connects seamlessly to traditional ideas such as the ideomotor principle, which has been part and parcel of psychological theorizing from the very early days of experimental psychology (James already referred to it). In fact there are many reviews on ideomotor theory (Shin et al., 2019 add others) also underlying the relevance of this conceptual framework. Effect anticipation plays a dominant role in action, motivation, and emotion literature. Therefore, this is in our view not 'niche'. Modern event-coding theories like TEC/BRAC are in the tradition of the ideomotor theory and give a particular perspective on action control. The important point is that TEC and especially BRAC can account for a vast number of action control processes for which more specific theories have been put forward. Since BRAC derives from ideomotor theory and is able to capture a much broader spectrum of action control processes than other theories (e.g. specific ones in task switching etc.), BRAC offers a broad scope that might eventually also make contact to other areas of the literature.

So, what did we do with this issue in the revision? We more clearly state, that we look at action control from a certain perspective (i.e. the ideomotor one), and that alternatives exist, of course. We also state the scope of our approach (for instance, decision making is not something we think we cover here). The separation of the main text and the boxes (with the definitions) is helping here, too. Finally, at the end of the text we reflect open questions/issues.

We do think that these measures give a more balanced view on these consensus-definitions and how they can be used. Again, we hope (and think) that researchers using action control tasks find these definitions helpful even if their research is not concerned with ideomotor theory explicitly.

Minor points:

In the definition of “event file”, the factor of time should be mentioned, in that it is the simultaneous or near-simultaneous experience of different stimulus features, etc. that results in them being bound together. Relatedly, I think a more explicit delineation of the current definition of “events” and the literature on event cognition (or event segmentation) would be useful somewhere in the paper. The latter literature is more concerned with temporally extended (real life) events, whereas the event coding literature seems to be focused on instantaneous, “trial-like” events exclusively (which is arguably further removed also from the everyday usage of “event”, so this might also be worth pointing out early on).

RESPONSE: Thank you. We have revised the definition of event file (incl. its decay etc.) and made it clearer. Also, we explicitly state the relation to event segmentation. As the event-file is the central concept we discuss it in the main text and not in the ‘boxes’.

In the same section, the distinction drawn between Logan’s instance theory and event coding is not very clear, please elaborate some more. (Specifically, “one event file each” sounds a lot like “one memory trace per instance”).

RESPONSE: We do not have the space to separate event-coding theory and the event-file concept with detail from Logan’s instances. Yet, at several places of the text differences are now hopefully clearer.

On line 169, the authors use (I believe for the first time) the term “feature code”, and later they discuss things like “code occupation”. The term “code”, as used in this literature, also needs a definition, in my view.

RESPONSE: Thank you, we revised these aspects in the manuscript. We now put several boxes (according to the Editor’s advice) to present clear definitions and relations with other concepts. Feature codes refer to the internal representation of features (e.g. of physical features of stimuli).

The “feature weighting” section seems to describe an attentional function (weighing task-relevant features more than irrelevant ones) without using the term “attention”. What is the relationship between feature weighting and attention?

RESPONSE: Thanks for this comment. Now we state that attention plays a role in feature weighting: “These weights are modulated by top-down factors, such as attention and task relevance, as well as bottom-up factors, such as salience.”

*In the action plan section, it is stated that an action plan precedes *every* action. This sounds odd when applied to completely reflexive or highly overlearned reactions. Is it really a “plan” when I pull away my hand from a hot stove plate?*

RESPONSE: We clarified it by adding “goal directed” before action in the following sentence:
“An action plan is a representation that precedes every goal-directed action”

On lines 381-383 the authors bring in the notion of event segmentation, but I don't think whatever point they are trying to make here is explained sufficiently.

RESPONSE: Thank. While revising the manuscript we put more emphasis on the distinction between event files, event segmentation and the role of context.

Line 388: "...as one brick...". I had trouble understanding this sentence – what does this mean? Is this meant to refer to a “building block”? Please clarify!

RESPONSE: This passage was deleted.

Comments to points raised by Reviewer 2

Reviewer #2 (Remarks to the Author):

Frings et al. present a short article attempting to provide consensus definitions of important terms in the action control field. The article is quite clear and will be of use to those in the field. It is highly focused on the action control field, and therefore may be of less interest to those outside the field however I believe it will still provide a useful contribution to the literature. I do have a few comments that the authors might find useful:

RESPONSE: Thank you for this positive feedback!

1. The brevity of the article sometimes means the authors presuppose knowledge, and therefore the writing can be slightly confusing. For example, in the "feature weightings" section you state "Such weights, in turn, can influence effects of code confusion in partial repetition effects". Sentences like this (of which there are several) might be better unpacked and explained a bit more clearly.

RESPONSE: We agree that the previous writing was too “packed”. We revised the entire manuscript and clarified connections between inter-related conceptual definitions.

2. In the "binary vs configural bindings" section, it is not clear to me whether there is a difference between multiple binary bindings vs a single configural binding. Are these two possibilities distinguishable?

RESPONSE: Thanks for this comment. The two possibilities are indeed difficult to distinguish empirically, yet they are theoretically different. We revised the binary vs configural bindings section to make this point clearer.

3. In the "action plan" section it wasn't clear to me whether an "action plan" was always online - an active representation created just before the action occurs - or whether it can also be stored/retained (i.e., within an event-file). A bit more background information/examples would help here.

RESPONSE: We improved the wording. Yet, unfortunately we do not have the space for examples.

4. Sometimes your "definitions" are more about the wider theory than a clear definition. For example, in the "event-file termination" section I don't think you actually define the term. It might be obvious, but given this article is about definitions this still seems important. A broader point is also perhaps the need to make clear what isn't included in each definition - how has the term been used differently in the past which you think was incorrect? This latter point might further help clear up confusion in the literature.

RESPONSE: The separation of main text and boxes hopefully cured this issue.

5. In the "event-file vs episode" section you link to episodic memory (as defined by Tulving), but don't discuss the "event segmentation" literature (e.g., Kurby & Zacks, 2008, TiCS). This seems relevant, given they specifically use the term "events" and it is more related to online processing of information than the episodic memory literature.

RESPONSE: We now refer to event-segmentation (in the sense of Zacks' approach) in the limitation section.

5. In the "retrieval" section, and earlier in the paper, you say that event-files are 'retrieved'. How is this distinguishable from continued activation of an event-file from the prime trial in the case of an immediate probe? Is retrieval always thought to occur rather than simple continued activation/maintenance?

RESPONSE: We do not have the space for discussing this – yet, in the supplement several papers are cited that are concerned with exactly this issue.

Comments to points raised by Reviewer 3

Reviewer #3 (Remarks to the Author):

This article by Fringe and colleagues is essentially a glossary of concepts that arise in the context of ideomotor-based theories of action control. As someone not initially well-versed in

these specific theories (for context, my research is focused on behavioral investigation and computational modeling of motor control, motor learning and cognitive control), I did find the article helpful to better understand the theory and current issues better. That said, it's certainly not a self-contained introduction to the theory. There is very little discussion of the evidence supporting these theories.

I am sympathetic to the importance of ensuring that all researchers reach a consensus on what different terms mean. A lot of unnecessary confusion and disagreement can arise from inconsistent terminology, leading to wasted effort, particularly when trying to bridge across research areas. With that in mind, I do have some comments about the terminology and definitions provided here, which I feel might end up creating barriers to other researchers engaging with these theories.

RESPONSE: Thank you for this positive feedback!

The main issue I have is that the terminology seems to blur the boundary between describing the phenomena that the theory seeks to explain, and describing the theory itself. Right from the beginning, on line 152, a "Response" (second definition) is defined as "a motor pattern that is cognitively represented by, and accessed through, its anticipated perceptual consequences." So, immediately, the definition here of what should be a very simple term ("response") – used for decades throughout neuroscience and psychology – has the ideomotor theory embedded into it. What would a motor pattern that is NOT represented in terms of its perceptual consequences be called? Do the authors believe that no such thing is possible? This creates exactly the kind of conceptual barrier across sub-disciplines that this article is seeking to overcome. A reader not already subscribed to the ideomotor principle will be alienated already.

There are more examples where this kind of issue arises. Another one is on line 169 - "a stimulus might instruct a specific response or can be the outcome of a response". This is fundamentally at odds with how the term "stimulus" is used elsewhere in neuroscience and psychology. If you are bought into the ideomotor principle, then this statement can make some sense. But I suspect this kind of definition, which again embeds the ideomotor principle is being embedded into the definition of terms, will create a barrier to integrating these ideas with other research areas.

A somewhat related issue is on line 305: "Binding effects in performance refer to longer response times in partial repetition conditions compared to either full repetitions or full change or both." My problem here is that a straightforward empirical phenomenon (differences in reaction time) is described by the putative underlying mechanism (binding). This will likely result in confusion (or worse, dogma) about how such effects can be explained.

In general, I would recommend that separate terminology should be used to describe

tasks/empirical phenomena versus possible mechanisms/interpretations. If the terminology used to describe the experimental paradigms and empirical phenomena cannot be disentangled from the theory that explains them then that is problematic. I suspect this type of issue will prove to be an obstacle to the insights and ideas here being appreciated widely and incorporated into a broader research program. I should emphasize that I think this are widespread issues in neuroscience/psychology (e.g. tasks like Stroop/Simon are often described as "inhibition tasks" – a putative mechanisms).

I want to emphasize that I offer these comments in the spirit of trying to help support the authors' initiative to improve consensus and sharing of ideas across sub-disciplines. I find the underlying ideas interesting and I sincerely hope that we can make progress towards overcoming the siloed nature of a lot of research (especially in the case of action control).

RESPONSE: We thank the reviewer for this comment. In principle, we agree with her/him. Yet, on the other hand, what we tried to do is to achieve consensus amongst researchers working on action control and acknowledging the ideomotor principle. In other words, some of our definitions (like the event-file) stem from ideomotor/event-file coding thinking – without these theories, these definitions do not make sense. So, a complete separation of theory and definitions is (i) not what we had intended actually, (ii) not what in our view the field is foremost needing, as there is already a lot of confusion with these terms even amongst researchers agreeing on event-coding perspectives, and (iii) for some definitions not possible. Moreover, we agree that some definitions (of e.g. 'stimulus' or 'response' that you mentioned) appear somewhat counterintuitive when viewed from established theoretical approaches. But eventually it is this sort of 'thought provocation' (ending in either agreement or disagreement) to which we want to invite readers.

So, how did we handle this issue in the revision?

We more clearly state, that we look at action control from an event-coding perspective, and explicitly state that alternatives exist. We also state the scope of our approach (for instance, decision making is not something we think we cover here).

Still, we tried to separate theory and definitions whenever possible or are more careful with our language –e.g. "partial repetition cost are interpreted as reflecting binding and retrieval". The separation of the main text and the boxes with the definitions is helping to do that.

As stated above, we do think that these measures give a more balanced view on these consensus-definitions and how they can be used. We hope (and think) that researchers using action control tasks find these definitions helpful even if their research is not concerned with ideomotor theory.

Finally, on a different note, I do have some concerns about the extent to which the BRAC theory can be falsified. Later sections of the paper refer to a number of effects that are difficult to reconcile with the basic theory, but these are invariably explained away by

amendments/extensions to the core theory. There is no discussion of alternative theories or perspectives. On that note, I wonder if some of the effects could be understood in terms of normative (e.g Bayesian) principles, rather than an ever-more-complex event-file mechanism? Note that such an approach (if it's possible; I have no idea) wouldn't be mutually exclusive to a mechanistic account, but might provide an alternative/complementary point of view through which some phenomena may be more easily explained.

RESPONSE: BRAC is a framework and thus it is hard to test/falsify this framework with a single experimentum crucis, that is true. Yet, hypotheses derived from BRAC have been extensively tested (and were falsified or confirmed) in the literature – still this article is not the place for an exhaustive discussion of these findings.

Yet, we do now discuss alternative theories, like predictive coding, the reviewer is pointing to.

28th Nov 23

Dear Christian

Your manuscript titled "Consensus definitions of Perception-Action-Integration in Action Control" has now been seen by our reviewers, whose comments appear below. In light of their advice I am delighted to say that we are happy, in principle, to publish a suitably revised version in Communications Psychology under the open access CC BY license (Creative Commons Attribution v4.0 International License).

We therefore invite you to revise your paper one last time to address the remaining editorial requests. At the same time we ask that you edit your manuscript to comply with our format requirements and to maximise the accessibility and therefore the impact of your work. To facilitate the task, I include an edited version of your manuscript.

EDITORIAL REQUESTS:

Please also undertake the deposition of your online library of relevant literature and link to the public version in the manuscript. This deposition should be truly public, i.e., accessible without login. We recommend a CC-BY license for the deposition.

SUBMISSION INFORMATION:

In order to accept your paper, we require the files listed at the end of the Editorial Requests Table; the list of required files is also available at <https://www.nature.com/documents/commsj-file-checklist.pdf> .

OPEN ACCESS:

Communications Psychology is a fully open access journal. Articles are made freely accessible on publication under a [CC BY](http://creativecommons.org/licenses/by/4.0) license (Creative Commons Attribution 4.0 International License). This license allows maximum dissemination and re-use of open access materials and is preferred by many research funding bodies.

For further information about article processing charges, open access funding, and advice and support from Nature Research, please visit <https://www.nature.com/commspsychol/article-processing-charges>

At acceptance, you will be provided with instructions for completing this CC BY license on behalf of all authors. This grants us the necessary permissions to publish your paper. Additionally, you will be asked to declare that all required third party permissions have been obtained, and to provide billing information in order to pay the article-processing charge (APC).

* **TRANSPARENT PEER REVIEW:** Communications Psychology uses a transparent peer review system. On author request, confidential information and data can be removed from the published reviewer

reports and rebuttal letters prior to publication. If you are concerned about the release of confidential data, please let us know specifically what information you would like to have removed. Please note that we cannot incorporate redactions for any other reasons.

[Link redacted]

Best regards,

Marike

Marike Schiffer, PhD
Chief Editor
Communications Psychology

REVIEWERS' COMMENTS:

Reviewer #1 (Remarks to the Author):

The authors did an excellent job of addressing my concerns, and I also think that the box structure has made the paper overall easier to digest. I'm sure it will serve as a useful reference to a growing field.

Reviewer #2 (Remarks to the Author):

The authors have responded well to the first round of reviews. I believe the manuscript is more targeted and concise now and will be a good contribution to the literature. I have no further comments.

Reviewer #3 (Remarks to the Author):

The authors have addressed most of the concerns I previously had. Their response also clarifies for me what their goals were in writing this article, and I think the article accomplishes that.

This article won't achieve much in the way of outreach to other scientific communities – more work in future will be needed for that. But as a concise glossary of terms for this kind of research it will likely be of value to those actively working in this area.